# Theoretical Validation and Hardware Implementation of Dynamic Adaptive Scheduling for Heterogeneous Systems on Chip [†]

A. Alper Goksoy [1,*] , Sahil Hassan [2], Anish Krishnakumar [1], Radu Marculescu [3] , Ali Akoglu [2] and Umit Y. Ogras [1]

1 Department of Electrical and Computer Engineering, University of Wisconsin-Madison, Madison, WI 53706, USA; anish.n.krishnakumar@wisc.edu (A.K.); uogras@wisc.edu (U.Y.O.)
2 Department of Electrical and Computer Engineering, University of Arizona, Tucson, AZ 85721, USA; sahilhassan@arizona.edu (S.H.); akoglu@arizona.edu (A.A.)
3 Department of Electrical and Computer Engineering, The University of Texas at Austin, Austin, TX 78712, USA; radum@utexas.edu
* Correspondence: agoksoy@wisc.edu
† This paper is an extended version of our paper published in IEEE Embedded Systems Letters (https://doi.org/10.1109/LES.2021.3110426).

**Abstract:** Domain-specific systems on chip (DSSoCs) aim to narrow the gap between general-purpose processors and application-specific designs. CPU clusters enable programmability, whereas hardware accelerators tailored to the target domain minimize task execution times and power consumption. Traditional operating system (OS) schedulers can diminish the potential of DSSoCs, as their execution times can be orders of magnitude larger than the task execution time. To address this problem, we propose a dynamic adaptive scheduling (DAS) framework that combines the advantages of a fast, low-overhead scheduler and a sophisticated, high-performance scheduler with a larger overhead. We present a novel runtime classifier that chooses the better scheduler type as a function of the system workload, leading to improved system performance and energy-delay product (EDP). Experiments with five real-world streaming applications indicate that DAS consistently outperforms fast, low-overhead, and slow, sophisticated schedulers. DAS achieves a $1.29\times$ speedup and a 45% lower EDP than the sophisticated scheduler under low data rates and a $1.28\times$ speedup and a 37% lower EDP than the fast scheduler when the workload complexity increases. Furthermore, we demonstrate that the superior performance of the DAS framework also applies to hardware platforms, with up to a 48% and 52% reduction in the execution time and EDP, respectively.

**Keywords:** domain-specific SoC; DSSoC; task scheduling; runtime classification; policy switching

## 1. Introduction

Heterogeneous systems on chip (SoCs) combine the energy efficiency and performance of custom designs with the flexibility benefits of general-purpose cores. Domain-specific SoCs (DSSoCs), a subset of heterogeneous SoCs, are emerging examples that integrate general-purpose compute elements and hardware accelerators that target the commonly encountered tasks (i.e., computational kernels) in the target domain [1–4]. For example, a DSSoC for autonomous driving incorporates computer vision and deep learning accelerators, whereas a DSSoC for a 5G/6G communication domain accelerates signal processing operations such as a fast Fourier transform (FFT). In addition, general-purpose CPU clusters and programmable accelerators, such as coarse-grained reconfigurable architectures (CGRA), offer alternative execution paths for a broader set of tasks besides improving flexibility [2,5].

In contrast to fixed-function designs, a critical distinction of DSSoCs is their ability to run multiple applications from the same domain [6]. When multiple applications run

concurrently, the number of ready tasks can exceed the capacity of the available accelerators, resulting in resource contention. This resource contention leads to a complex runtime scheduling problem since there are many different ways to prioritize and run such tasks. For example, waiting for the most suitable resource to become available can lead to higher energy efficiency than resorting to an immediately available less suitable resource like a CPU core or a reconfigurable accelerator. Besides the complex runtime scheduling problem, recent studies show that execution times for applications of domain-specific systems are on the nanosecond scale [1,7]. Therefore, the classical scheduling problem encounters a new challenge in heterogeneous DSSoCs because domain-specific tasks can run in the order of nanoseconds, i.e., at least two or three orders of magnitude faster than general-purpose cores when they are executed on their specialized pipelines. If the scheduler takes a significantly longer amount of time to make a decision, it can undermine the benefits of hardware acceleration. For instance, the Linux Completely Fair Scheduler (CFS) takes 1.2 µs to make a scheduling decision when running on an Arm Cortex-A53 core [8–11]. This overhead is clearly unacceptable when there are many tasks with orders of magnitude faster execution times.

DSSoCs require fast scheduling algorithms to keep up with tasks that can run in the order of nanoseconds and achieve high efficiency. However, the poor scheduling decisions of simple low-overhead schedulers can decrease system performance, especially under heavy workloads. For example, Figure 1 shows the average execution time and energy-delay product for a workload mix of a wireless communication system. When the data rate is low (i.e., few frames are present concurrently), there is lower contention for the same SoC resources. Hence, a fast low-overhead scheduler (in this work, a lookup table) outperforms a more sophisticated scheduler (e.g., a complex heuristic or an integer programming-based scheduler) since it can make the same or very similar assignments with a significantly lower overhead. As the data rate increases, the number of concurrent frames, and hence the complexity of scheduling decisions, also grow. At the same time, the waiting times of the tasks in the core queues exceed their actual execution time in the accelerators. Therefore, a sophisticated scheduler outperforms a simple one (point A in Figure 1a), and the overhead of making better scheduling decisions pays off. In contrast, when an application involves a high degree of task-level parallelism, we expect more tasks in the ready queue waiting for a scheduling decision. For such applications, the scheduling overhead can dominate the execution time, even for lower data rates (Figure 1b). For these cases, using a sophisticated scheduler cannot outperform the scheduling decisions of a simple scheduler. As the data rate increases, a sophisticated scheduler can degrade system performance because of its dependence on the number of ready tasks. Therefore, a simple scheduler outperforms a sophisticated scheduler under various data rates. Hence, the trade-off between the scheduling decision quality and the scheduling overhead is an opportunity for exploitation.

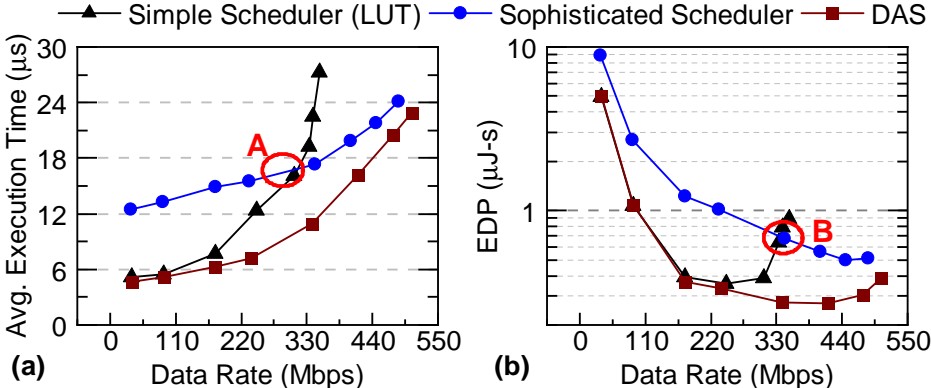

**Figure 1.** An example of the relationship of the (**a**) execution time and (**b**) energy-delay product (EDP) between simple low-overhead (lookup table or LUT) and sophisticated high-overhead schedulers.

To exploit the trade-off between the scheduler overhead and decision quality, we propose a *dynamic adaptive scheduling* (DAS) framework that combines the benefits of sophisticated and simple low-overhead scheduling algorithms using an integrated support mechanism. Making a scheduling decision at the scale of nanoseconds is highly challenging because executing possibly complex decisions and loading the necessary context, such as performance counters, requires extra time. Hence, even the data movement part alone can violate the fast decision target. Moreover, the framework needs to determine at runtime whether the low-overhead or sophisticated scheduler should run. The DAS framework outperforms both underlying schedulers despite these challenges by following *key observations* in the design process. *First,* the scheduling will be called with 100% certainty and use features (a subset of available performance counters). Therefore, prefetching the required features in a background process and writing them to a pre-allocated local memory location can hide the extra overhead. *Second,* the choice of a sophisticated or simple scheduler can be made by the same prefetching process before the subsequent scheduling process begins. If the simple scheduler is chosen, the only extra overhead on the critical path is the time to access the LUT, which is measured as 6 ns on the Arm Cortex-A53. We run the sophisticated scheduler at runtime only if a more complex scheduler is needed.

A preliminary version of this work appeared in the *IEEE Embedded Systems Letters* [12], where we presented the concept of dynamic adaptive scheduling using high-level event-driven simulations. It showed that adaptively using a slow and a fast scheduler can enable better performance and energy efficiency than any of the schedulers alone. However, the preliminary work relied on high-level simulations without theoretical grounds and experimental evaluations. This paper significantly expands the preliminary work and makes the following novel contributions:

- Theoretical proof of the DAS framework and its experimental validation using a DSSoC simulator;
- Integration of the DAS framework with an open-source runtime environment and its training and deployment on a Xilinx Zynq ZCU102 SoC;
- Extensive performance evaluation in the trade space of the execution time, energy, and scheduling overhead over the Xilinx Zynq ZCU102 SoC based on workload scenarios composed of real-life applications.

The rest of this paper is organized as follows. Section 2 reviews the related works. We describe the proposed DAS framework and algorithms in Section 3. Section 4 discusses and analyzes the experimental results on a DSSoC simulator with real-world applications, whereas Section 5 presents the training and implementation details of the DAS framework on an FPGA emulation environment using real-world applications. Finally, Section 6 concludes this paper.

## 2. Related Works

Schedulers have evolved significantly to adapt to different requirements and optimization objectives. Static [13,14] and dynamic [10,15–17] task scheduling algorithms have been proposed in the literature. Completely Fair Scheduler (CFS) [10] is a dynamic approach that is widely used in Linux-based OSs and aims to provide resource fairness to all processes, whereas the static approaches presented in [13,14] optimize the makespan of applications. CFS [10] was initially developed for homogeneous platforms but it can also handle heterogeneous architectures (e.g., Arm big.LITTLE). While CFS may be effective for client and small-server systems, high-performance computing (HPC) and high-throughput computing (HTC) necessitate different scheduling policies. These policies, such as Slurm and HTCondor, are specifically designed to manage a large number of parallel jobs and meet high-throughput requirements [18,19]. On the other hand, DSSoCs demand a novel suite of efficient schedulers that execute at nanosecond-scale overheads since they deal with scheduling tasks that can execute in the order of nanoseconds.

The scheduling overhead problem and scheduler complexities are discussed in [20–27]. The authors of [20] propose two dynamic schedulers known as CATS and CPATH, where

CATS detects the longest path and CPATH detects the critical paths in the application. The CPATH algorithm shows inefficiency in terms of its higher scheduling overhead. Motivated by high scheduling overheads, Anish et al. [9] propose a new scheduler that approximates an expensive heuristic algorithm using imitation learning with low overhead. However, the scheduling overhead is still approximately 1.1 μs, making it impractical for DSSoCs with nanosecond-scale task execution. The authors of [27] propose a neural network-based scheduler selector that schedules eight independent tasks to eight cores from three different architectures. However, using a neural network-based architecture for the selector results in high overheads unsuitable for DSSoCs. Energy-aware schedulers for heterogeneous SoCs have limited applicability to DSSoCs because of their complexity and large overheads [28–31]. There are other papers that discuss the overhead of data offloading through dispatching between accelerators and CPU units using schedulers [32,33].

Several scheduling algorithms that demonstrate the benefits of using multiple schedulers are proposed in [34–36]. Specifically, the authors of [34] propose a technique that dynamically switches between three schedulers to adapt to varying job characteristics. However, the overheads of switching between policies are not considered as part of the scheduling overhead. The approach in [36] integrates static and dynamic schedulers to exploit both design-time and runtime characteristics for homogeneous processors. The hybrid scheduler in [36] uses a heuristic list-based schedule as a starting point and then improves it using genetic algorithms. However, it does not consider the scheduling overheads of the individual schedulers. The authors of [21] discuss a performance comparison of a simple round-robin scheduler and a complex earliest deadline first (EDF) scheduler and their applicability under different system load scenarios.

Using insights from the literature, we propose a novel scheduler that combines the benefits of the low scheduling overhead of a simple scheduler and the decision quality of a sophisticated scheduler (described in Section 3.3) based on the system workload intensity. To the best of our knowledge, this is the first approach that uses a novel runtime preselection classifier to choose between simple and sophisticated schedulers at runtime to enable scheduling with low-energy and nanosecond-scale overheads in DSSoCs.

## 3. Dynamic Adaptive Scheduling Framework

In this section, we start by introducing the preliminaries used in the proposed DAS approach (Section 3.1) and then present the dataset generation process and the online usage of DAS models (Section 3.2). Finally, we present the selection of fast and sophisticated schedulers in Section 3.3.

### 3.1. Overview and Preliminaries

This work considers streaming applications that can be modeled by data flow graphs (DFGs). More precisely, consecutive data frames are pipelined through the tasks in the graph. The current practice of scheduling is limited to a single scheduler policy. On the other hand, DAS allows the OS to choose one scheduling policy $\pi \in \mathbf{\Pi}_S = \{F, S\}$, where $F$ and $S$ refer to the *fast* and *slow* (*or sophisticated*) schedulers, respectively. When the task becomes ready (predecessors of the task are completed), the OS can call either a slow ($\pi = S$) or a fast ($\pi = F$) scheduler as a function of the workload and system state. The OS collects a set of performance counters during the execution to enable two aspects of the DAS framework: (1) precise assessment of the system state, and (2) desirable features for the classifier to switch between the *fast* and *slow* schedulers at runtime. Table 1 shows the various types of performance counters that are collected for the DAS framework. The total number of performance counters is 62 for a DSSoC with 19 PEs. The goal of the slow scheduler $S$ is to handle more complex scenarios when the task wait times dominate the execution times. In contrast, the fast scheduler $F$ aims to approach the theoretically minimum (i.e., zero) scheduling overhead by making decisions in a few cycles with a minimum number of operations. *The DAS framework aims to outperform both underlying schedulers by dynamically switching between them as a function of the system state and workload.*

**Table 1.** Type of performance counters used by DAS for system state representation and runtime classification between schedulers.

| Type | Features |
|---|---|
| Task | Task ID, Execution time, Power consumption, Depth of task in DFG, Application ID, Predecessor task ID and cluster IDs, Application type |
| Processing Element (PE) | Earliest time when PE is ready to execute, Earliest availability time of each cluster, PE utilization, Communication cost |
| System | Input data rate |

*3.2. DAS Preselection Classifier*

The first runtime decision of DAS is the selection of a fast or slow scheduler. We should optimize this decision to approach the goal of zero scheduling overhead, as it is on the critical path of both schedulers. One of the novel contributions of DAS is that we recognize this selection as a deterministic task that will be eventually executed with a probability of one. Therefore, we prefetch the relevant features that are required for this decision to a pre-allocated local memory. We re-use a subset of the performance counters shown in Table 1 as desirable features to minimize the scheduling overhead. The relevant subset of features is presented in Section 4.2. To reflect the system state at that point in time, the OS periodically refreshes the performance counters. The DAS framework runs a lightweight classifier that determines whether the fast or slow scheduler should be used for the next ready task each time the features are refreshed. As it is refreshed with the features that reflect the most recent system state, we note that this decision will always be up to date. Thus, DAS can determine which scheduler should be called, even before a task is ready for scheduling. Consequently, the preselection classifier of the DAS framework introduces zero latency and minimal energy overhead. Next, we discuss the offline preselection classifier and its online use. The latter will cover the overhead involved in switching between policies.

**Offline Classifier Design:** The first step in the design process of the preselection classifier is to generate the training data based on the domain applications known at design time. Each scenario in the training data consists of concurrent applications and their respective data rates. For example, a combination of WiFi transmitter and receiver chains at a specific upload and download speed could be one such scenario. To this end, we run each scenario twice on an instrumented hardware platform or a simulator (see Figure 2).

*First Execution*: The instrumentation enables us to run *both fast and slow schedulers* each time a task scheduling decision is made. We note that the scheduler is invoked whenever a task is to be scheduled. If the decisions of the fast ($D_F$) and slow ($D_S$) schedulers for task $T_i$ are identical, then we label task $T_i$ with $F$ (i.e., the fast scheduler) and store a snapshot of the performance counters. The label $F$ implies that the fast scheduler can reach the same scheduling decision as the sophisticated algorithm under the system states captured by the performance counters. If the decisions of the schedulers are different, the label is left as *pending* and the execution continues by following the decision of the fast scheduler, as described in Figure 2. At the end of the execution, the training data contains a mix of both labeled ($F$) and pending decisions.

*Second Execution*: The same scenario is executed for the second time. This time, the execution always follows the decisions of the slow scheduler. At the end of the execution, we analyze the target metric, such as the average execution time or the total energy consumption. If a better result can be achieved using the slow scheduler, the pending labels are replaced with $S$ to indicate that the slow scheduler is preferred despite its larger overhead. Otherwise, we conclude that the lower overhead of the fast scheduler pays off, and the pending labels are replaced with $F$. An alternative approach is to evaluate each pending instance

individually; however, this would not offer a scalable solution, as scheduling is a sequential decision-making problem, and a decision at time $t_k$ affects the remaining execution.

**Figure 2.** Flowchart describing the construction of an Oracle to dynamically choose the best-performing scheduler at runtime.

In this work, the training data are generated using 40 distinct workloads. Each workload is a mix of multiple instances of five applications, consisting of approximately 140,000 tasks in total and executed at 14 different data rates, as detailed in Section 4.1 and Appendix B. A higher data rate presents a larger number of concurrent applications contending for the same SoC resources. Then, we design a low-overhead classifier using machine learning techniques and feature selection methods [37], as described in Section 4.2.

**Online Use of the Classifier:** The last step for the classifier is the deployment at runtime (last block in Figure 2). At runtime, a background process periodically updates a pre-allocated local memory with the subset of performance counters that the classifier requires for the scheduler selection. After each update, the classifier determines whether the fast $F$ or slow $S$ scheduler should be used for the next available task. When a new ready task becomes available, the features are already loaded and we know which scheduler is a better choice for the read task. Therefore, the DAS framework does not incur any extra delays on the critical path. Moreover, it has a negligible energy overhead, as demonstrated in Section 4, which is critical for the performance and applicability of such an algorithm.

### 3.3. Fast and Slow (F and S) Scheduler Selection

The proposed DAS framework can work with any choice of fast and slow scheduling algorithms. This work uses a lookup table (LUT) implementation as the fast scheduler, as the goal of the fast scheduler is to achieve almost zero overhead. The LUT stores the most energy-efficient processor in the target system as a decision for each known task in the target domain. Unknown tasks are mapped to the next available CPU core. Hence, the LUT access at runtime is the only extra delay on the critical path and overhead of the fast scheduler. To profile the scheduling overhead of the LUT, we developed an optimized C implementation with inline assembly code. We ran the script 10,000 times and averaged the latency and energy consumption. We used the on-chip TI INA226 power sensors to measure the energy consumption. Our experiments show that the fast scheduler takes $\sim$7.2 cycles (6 ns on an Arm Cortex-A53 at 1.2 GHz) on average and incurs a negligible (2.3 nJ) energy overhead.

The DAS framework uses a commonly used heuristic (earliest task first (ETF)) as the slow scheduler [38]. ETF is chosen since it recursively iterates over the ready queue tasks and processors to provide the schedule that achieves the fastest finish time, as shown in

Algorithm 1. It performs a comprehensive search, which can make the best decision when the system is loaded with many tasks. It starts from the first task $T_i$ in the ready queue $\mathcal{T}$. It computes the finish time of task $T_i$ on each PE $p_j$. Then, it continues this process for every task-PE pair ($T_i$-$p_j$). Then, it selects the task-PE pair (let us say $T'$-$p'$) that gives the earliest finish time. After assigning task $T'$ to PE-$p'$, the process repeats with the remaining tasks in the ready queue $\mathcal{T}$. Hence, its computational complexity is quadratic with respect to the number of ready tasks. ETF's detailed computational and energy overheads are presented in Section 4.1.

---

**Algorithm 1** ETF Scheduler

---

 1: **while** ready queue $\mathcal{T}$ is not empty **do**
 2:     **for** task $T_i \in \mathcal{T}$ **do**
 3:         /* $\mathcal{P}$ **= set of PEs** */
 4:         **for** PE $p_j \in \mathcal{P}$ **do**
 5:             $FT_{T_i,p_j}$ = Compute the finish time of $T_i$ on $p_j$
 6:         **end for**
 7:     **end for**
 8:     ($T'$, $p'$) = Find the task and PE pair that has the minimum FT
 9:     Assign task $T'$ to PE $p'$
10: **end while**

---

The detailed theoretical proof that shows that DAS achieves superior performance compared to the baseline schedulers and its experimental validation are presented in Appendix A.

## 4. Evaluation of DAS Using Simulations

Section 4.1 describes the experimental setup used in this work. Section 4.2 explores different machine learning methods and features of DAS. The evaluation and detailed analysis of DAS for different workloads are shown in Section 4.3. Finally, we demonstrate the latency and energy consumption overheads of DAS in Section 4.4.

### 4.1. Simulation Setup

This section presents the setup used for profiling the scheduling overhead of complex and basic schedulers, the simulation framework that has been used for performance analysis and training data generation, and the SoC configuration that we used.

**Domain Applications:** The DAS framework is evaluated using five real-world streaming applications: range detection, temporal mitigation, WiFi-Transmitter, WiFi-Receiver applications, and a proprietary industrial application (App-1), as summarized in Table 2. We then construct 40 different workloads for runtime analysis of the schedulers used in this paper. All workloads are run in streaming mode, and for each data point in the figures in Section 4.3, approximately 10,000 tasks are scheduled. Details of the workload mixes are given in Appendix B.

**Emulation Environment:** Conducting a realistic runtime overhead and energy analysis is one of our key goals in this study. For this purpose, we leverage a Compiler-integrated, Extensible DSSoC Runtime (CEDR) framework introduced by Mack et al. [5]. The CEDR has been validated extensively on x86 and Arm-based platforms. It enables pre-silicon performance evaluations of heterogeneous hardware configurations composed of mixtures of CPU cores and accelerators based on dynamically arriving workload scenarios. Compared to other emulation frameworks (e.g., ZeBu [39] and Veloce [40]), this portable and open-source environment offers distinct plug-and-play integration points, where developers can individually integrate and evaluate their applications, scheduling heuristics, and accelerator IPs in a realistic system.

**Table 2.** Characteristics of applications from radar processing and wireless communication domains used in this study. FFT = fast Fourier transform; FEC = forward error correction; FIR = finite impulse response; SAP = systolic array processor.

| Application | Number of Tasks | Supported Clusters |
|:---:|:---:|:---:|
| Range Detection | 7 | big, LITTLE, FFT, SAP |
| Temporal Mitigation | 10 | big, LITTLE, FIR, SAP |
| WiFi-TX | 27 | big, LITTLE, FFT, SAP |
| WiFi-RX | 34 | big, LITTLE, FFT, FEC, FIR, SAP |
| App-1 | 10 | LITTLE, FIR, SAP |

The emulation framework [41] combines the compile-time analysis with the runtime system. The compilation process involves converting each application to LLVM intermediate representation (IR), identifying which section of the code should be labeled as "kernels" (frequently executing IR-level blocks) or "non-kernels", and automatically refactoring the LLVM IR into a directed acyclic graph (DAG), where each node represents a kernel. Abstracting the application with a DAG, compile-time flow generates a flexible binary structure for the runtime system to be able to invoke each function call on all its supported processing elements (PEs) in the target architecture. The runtime system monitors the state of system resources, dynamically parses the arriving applications, generates the queue of tasks that have resolved dependencies (called ready queue), schedules the ready-queue tasks based on the user-defined scheduling policy, and manages the data transfers to and from the PEs. The runtime system supports scheduling heuristics such as round robin, earliest finish time (EFT), and earliest task first (ETF). We selected the ETF scheduler as the complex scheduler of our analysis because of its complexity and the quality of decisions it provides. As shown in Algorithm 1, ETF recursively iterates over the ready-queue tasks and PEs to find the task-to-PE mapping decisions that result in the minimum finish time. It carries out this process recursively until each ready task is scheduled to a PE.

We generate a wide range of workloads, ranging from all application instances belonging to a single application to a uniform distribution of all five applications, to evaluate the impact of the scheduling overhead of the ETF scheduler. Details about these workload mixes are provided in Appendix B. We illustrate the impact of the number of tasks in the ready queue on the scheduling overhead of ETF in Figure 3, as measured on the Xilinx Zynq ZCU102 [42]. The x-axis in the plot shows the number of tasks that are in the ready queue for each workload scenario, and the y-axis shows the scheduling overhead of the ETF scheduler. We generate a quadratic equation to formulate the ETF scheduling overhead observed at runtime based on these measurements. Later, we utilize this equation to evaluate the average execution time and EDP of the DAS scheduler on the simulator. We note that the DAS framework is by no means limited to the processors, schedulers, and applications that are used for demonstration purposes.

**Simulation Environment:** We use DS3 [43], an open-source domain-specific system-on-chip simulation framework, to perform detailed evaluations of our proposed scheduling approach. DS3 is a high-level simulation tool that includes built-in scheduling algorithms, models for PEs, interconnect, and memory systems. The framework has been validated on Xilinx Zynq ZCU102 and Odroid-XU3 (with a Samsung Exynos 5422 SoC) platforms. Therefore, it is a robust and reliable tool for performing a detailed exploration and evaluation of our proposed scheduling approach. DS3 supports the execution of streaming applications (represented as directed flow graphs). It allows multiple modes for injecting new applications at runtime, including fixed-interval injection and exponential distribution-based job injection. The inputs to the tool are the execution profiles of the processing elements in the SoC, application DFGs, and the interconnect configuration. At the end of the simulation, DS3 provides various workload statistics, including the injection rate,

execution time, throughput, and PE utilization. We use these metrics provided by DS3 in our extensive evaluation to demonstrate the benefits of the DAS scheduling methodology. **DSSoC Configuration:** We construct a DSSoC configuration that comprises clusters of general-purpose cores and hardware accelerators. The hardware accelerators include fixed-function designs, and a multi-function systolic array processor (SAP). The application domains used in this study are wireless communications and radar systems, and hence, accelerators that expedite the execution of relevant tasks are included.

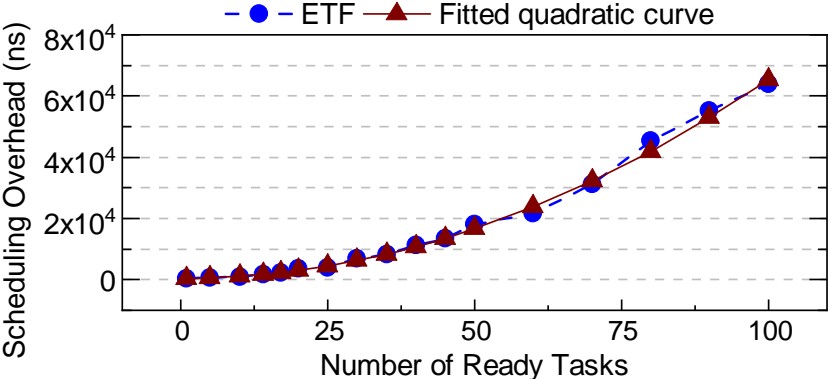

**Figure 3.** ETF scheduling overhead and fitted quadratic curve.

The DSSoC used in our experiments includes the Arm big.LITTLE architecture, each with four cores. We also include dedicated accelerators for fast Fourier transform (FFT), forward error correction (FEC), finite impulse response (FIR), and SAP. We include four cores each for the FFT and FIR accelerators, one core for the FEC, and two cores for the SAP. The FEC accelerator accelerates the execution of encoder and decoder operations, whereas the SAP accelerator accelerates multiple tasks for the application domain. In total, the DSSoC integrates 19 PEs with a mesh-based network on chip (NoC) to enable efficient on-chip data movement. This configuration is summarized in Table 3.

**Table 3.** DSSoC configuration used for DAS evaluation.

| Processing Cluster | No. of Cores | Functionality |
| --- | --- | --- |
| LITTLE | 4 | General purpose |
| big | 4 | General purpose |
| FFT | 4 | Acceleration of FFT |
| FEC | 1 | Acceleration of encoding and decoding operations |
| FIR | 4 | Acceleration of FIR |
| SAP | 2 | Multi-function acceleration |
| TOTAL | 19 | |

**Performance Metrics:** We use the average execution time per application instance as our main performance metric. This metric is calculated by dividing the time each application instance is executed by the number of application instances. Another metric used for performance comparison is the average job slowdown. It is calculated as the average execution time of a scheduler divided by the average execution time of the idealized version of the ETF scheduler to show the distance of scheduler performances from an ideal scenario. Furthermore, we use the energy-delay product (EDP) metric to analyze the effects on energy consumption. The EDP is calculated as the multiplication of the total execution time (makespan) and energy consumption.

*4.2. Exploration of Machine Learning Techniques and Feature Space for DAS*

We train DAS models using various machine learning methods. As we target a model with a very low scheduling overhead, our analysis of the method selection considers

classification accuracy and model size as the main metrics. Specifically, we investigated support vector classifiers (SVC), decision tree (DT), multi-layer perceptron (MLP), and logistic regression (LR). The training process with SVCs with simple kernels exceeded 24 h, rendering it infeasible. The latency and storage requirements of the MLP (one hidden layer and 16 neurons) did not fit the budget of low-overhead requirements. Therefore, these two techniques were excluded from the rest of the analysis. Table 4 summarizes the classification accuracy and storage overheads for the logistic regression and decision tree classifiers as a function of the number of features and the depth of the tree for the decision trees.

**Table 4.** Classification accuracies and storage overheads of DAS models with different machine learning classifiers and features.

| Classifier | Tree Depth | Number of Features | Classification Accuracy (%) | Storage (KB) |
|:---:|:---:|:---:|:---:|:---:|
| LR | - | 2 | 79.23 | 0.01 |
| LR | - | 62 | 83.1 | 0.24 |
| DT | 2 | 1 | 63.66 | 0.01 |
| DT | 2 | 2 | 85.48 | 0.01 |
| DT | 3 | 6 | 85.51 | 0.03 |
| DT | 2 | 62 | 85.9 | 0.01 |
| DT | 16 | 62 | 91.65 | 256 |

**Machine Learning Technique Exploration:** The DT classifiers achieved similar or higher accuracies compared to the LR classifiers with lower storage overheads. While a DT with a depth of 16, which uses all the features, achieved the best classification accuracy, there was a significant impact on the storage overhead, which, in turn, affected the latency and energy consumption of the classifier. In comparison, DTs with tree depths of 2 and 4 had negligible storage overheads with competitive accuracies (>85%). Hence, we adopted the DT classifier with a depth of 2 for the DAS framework.

**Feature Space Exploration:** We collected 62 performance counters in our training data. Selecting a subset of these counters as the DAS classifier features is crucial for minimizing the energy and performance overheads. A systematic feature space exploration was performed using feature selection and importance methods such as analysis of variance (ANOVA), F-value, and Chi-squared statistics [37]. Among the top six features, increasing the feature list from a single feature (*input data rate*) to two features with the addition of the *earliest availability time of the Arm big cluster* increased the accuracy from 63.66% to 85.48%. We used an 8-entry × 16-bit shift register to track the data rate at runtime. Therefore, we selected the two most important features: data rate and the earliest availability time of the Arm big cluster to design the DAS classifier model with a decision tree of depth 2. We note that these two features do not contain task-related information. Hence, this enables DAS to be compatible with diverse task scenarios without incurring additional overheads since it is not on the critical path.

*4.3. Performance Analysis for Different Workloads*

This section compares the DAS framework with the LUT (fast), ETF (sophisticated), and ETF-ideal schedulers. ETF-ideal is a version of the ETF scheduler that ignores the scheduling overhead component. Therefore, ETF-ideal is an idealized version of a sophisticated scheduler that helps us establish the theoretical upper bound of the achievable execution time and EDP. Out of the 40 workloads described in Section 3.2, 3 representative workloads were chosen for a detailed analysis of the execution time and EDP trends. The three chosen workloads exhibit different data rates, which are a function of the applications in the workload. Workload 1 (Figure 4a,d) exhibits a low data rate (complex applications), Workload 2 (Figure 4b,e) exhibits moderate data rates, and Workload 3 (Figure 4c,f) exhibits a high data rate (simplest applications).

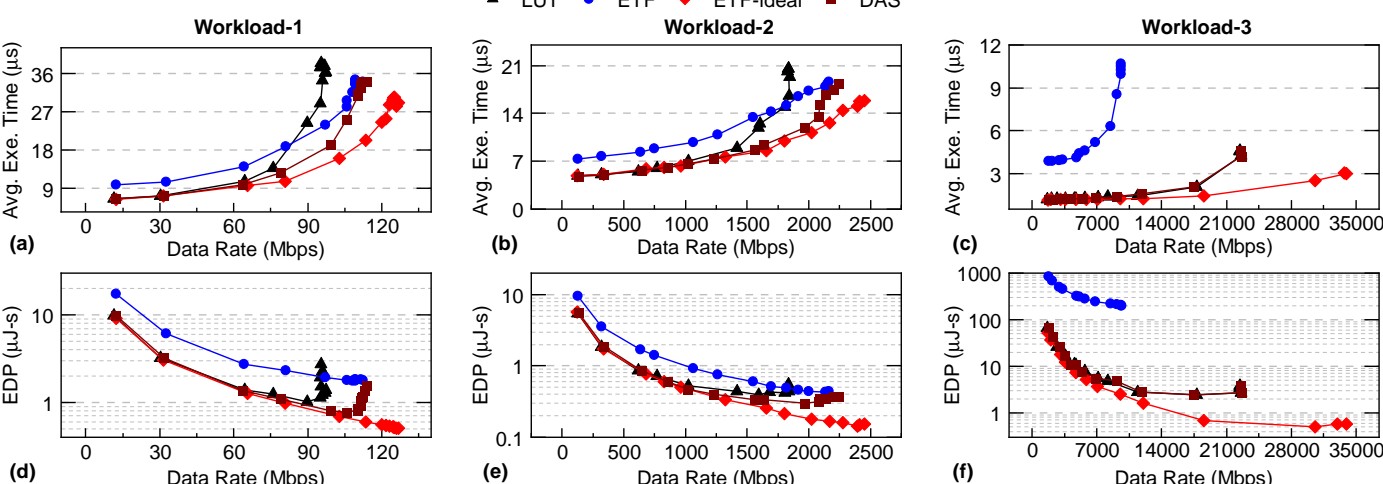

**Figure 4.** A comparison of the average execution time (**a**–**c**) and energy-delay product (EDP) (**d**–**f**) between DAS, the lookup table (LUT), earliest task first (ETF), and ETF-ideal for three different workloads.

Figure 4a–c shows a comparison of the execution times of the DAS, LUT, ETF, and ETF-ideal schedulers for three different workloads, whereas Figure 4d–f show their corresponding EDP trends. For Workload 1 and Workload 2, the system is not congested at low data rates. Hence, the performance of DAS is similar to that of the LUT, as expected. As the data rate increases, DAS aptly chooses between the LUT and ETF at runtime to achieve an execution time and EDP that is 14% and 15% better than the LUT, and 15% and 42% better than the ETF when used individually. For Workload 3, the execution time and EDP of the LUT are significantly lower than those of the ETF. DAS chooses the LUT for >99% of the decisions and closely follows its execution time and EDP. These results successfully demonstrate that DAS adapts to the workloads at runtime and aptly chooses between the LUT and ETF schedulers to achieve a low execution time and EDP.

The same study was extended to all 40 different workloads. DAS *consistently outperformed* both the LUT and ETF schedulers when they were used individually. At low data rates, i.e., when the LUT was better than the ETF, the DAS framework achieved more than a 1.29× speed up and a 45% lower EDP compared to the ETF while outperforming the LUT scheduler. Moreover, the DAS framework achieved as much as a 1.28× speed up and a 37% lower EDP than the LUT when the workload complexity increased. In summary, DAS was always significantly better than either one of the underlying schedulers.

Figure 5 summarizes the impact of the change in the workload composition on the performance of DAS using 25 workloads. The first three workloads are App-1-intensive, Workloads 4 to 8 are WiFi-intensive, and the last 11 are different combinations of applications. The average job slowdown was normalized to the ETF-ideal scheduler results. The plot shows that DAS performed better than the LUT and ETF schedulers for these different scenarios, bringing the slowdown to 1. This shows that DAS moved the execution time closer to the ideal scenario where the overhead of the scheduler was zero.

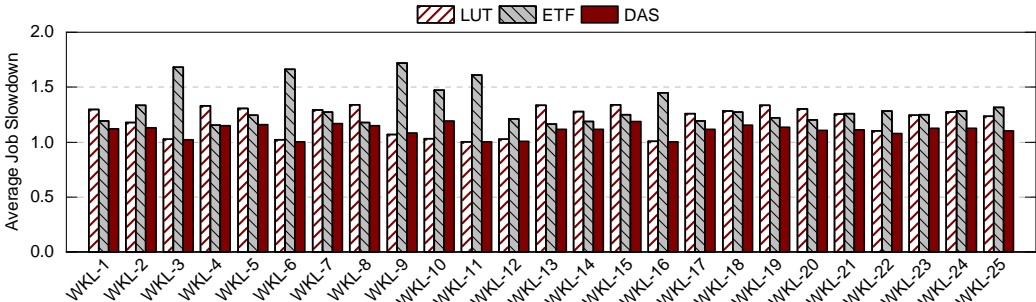

**Figure 5.** A comparison of the average job slowdown of DAS, LUT, and ETF for twenty-five workloads.

We also compared DAS with a heuristic approach, which selected the fast scheduler when the data rate was lower than a predetermined threshold and the slow scheduler when the data rate was higher than the threshold. The predetermined threshold was selected based on analyzing the training data used for DAS. The simulation results for this comparison for Workload 2 in Figure 4 are shown in Figure 6. The results show that the heuristic approach closely mimicked the behavior of the LUT and ETF schedulers below and above the threshold, respectively, without exhibiting intelligent adaptability. In contrast, DAS outperformed both schedulers, achieving a 13% reduction in the execution time compared to the heuristic scheduler.

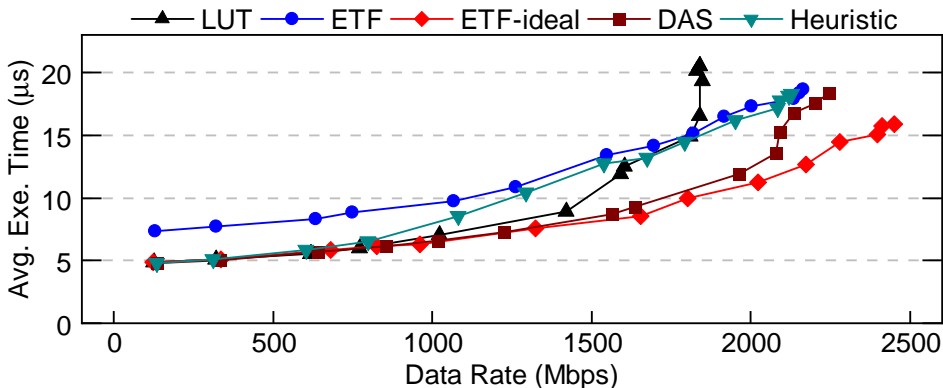

**Figure 6.** A comparison of the average execution times of DAS, LUT, ETF, ETF-ideal, and the heuristic approach.

### 4.4. Scheduling Overhead and Energy Consumption Analysis

This section analyzes the runtime selections made by the DAS framework between the LUT and ETF schedulers. Figure 7 plots the decision distribution of DAS for a workload that comprised all five applications on the primary axis. As a reminder, the low-overhead fast scheduler (LUT) performed well under a low system load. In contrast, the sophisticated scheduler (ETF) was superior under a heavy system load, achieving better performance and EDP. We note that DAS exploited the benefits of the LUT at low system loads and the ETF scheduler under heavy loads. As the data rate increased, the DAS framework judiciously utilized the ETF scheduler more frequently, as observed in Figure 7. Specifically, the DAS framework used the LUT scheduler for *all* decisions at a low data rate of 135 Mbps and the ETF scheduler for 95% of the decisions at the highest data rate of 2098 Mbps. At a moderate workload of 1352 Mbps, the DAS framework still used the LUT scheduler for 96% of the decisions. As a result, the average scheduling latency overhead of the DAS framework for all workloads was 65 ns.

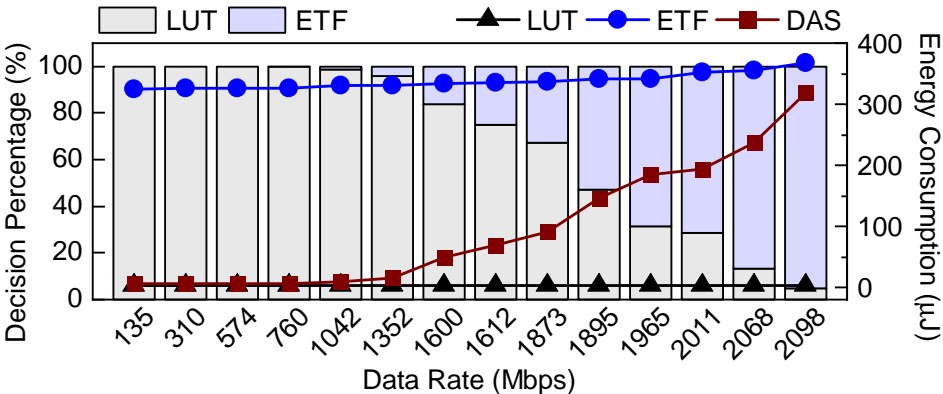

**Figure 7.** Decisions made by the DAS framework represented as bar plots and total scheduling energy overheads of LUT, ETF, and DAS as line plots.

The secondary axis in Figure 7 shows the total energy overhead of using different schedulers. As DAS used the LUT and ETF approaches based on the system load, its energy consumption varied from the LUT to the ETF. The energy consumption of the DAS was slightly higher than that of the LUT for low data rates because the DAS preselection model itself consumed a small amount of energy. On average, for all workloads, the energy overhead of DAS per scheduling decision was 27.2 nJ.

## 5. Evaluation of DAS Using FPGA Emulation

Section 4 presented detailed evaluations of the proposed DAS scheduler in a system-level simulation framework. This section focuses on its implementation on a real hardware platform that includes (1) heterogeneous PEs comprising general-purpose cores and hardware accelerators, and (2) a runtime framework that implements domain applications on heterogeneous SoCs and allows integrating customized schedulers. Section 5.1 first describes the SoC configuration, followed by an overview of the runtime environment and the data generation setup to train DAS models and their deployment in the runtime framework. Finally, Section 5.2 presents the performance evaluations of the proposed DAS scheduler on a hardware platform.

### 5.1. Experimental Setup

**DSSoC Configuration:** The domain applications presented in Section 4.1 frequently perform FFT and matrix multiplication operations. To this end, we constructed a hardware platform comprising hardware accelerators for FFT and matrix multiplication. Additionally, we included three general-purpose cores to execute the other tasks. The full-system hardware that integrated the cores and accelerators was implemented on a Xilinx Zynq UltraScale+ ZCU102 FPGA [42].

**Runtime Environment:** This study utilized the CEDR runtime framework [5] to implement DAS and evaluate its benefits for a DSSoC. CEDR allows users to compile and execute applications on heterogeneous SoC architectures. Furthermore, CEDR launches the execution of workloads comprising a combination of applications, each streaming with user-specified injection intervals. It offers a suite of schedulers and allows users to plug and play custom scheduling algorithms, making it a highly suitable environment for evaluating DAS. Therefore, we integrated DAS into CEDR and executed the workloads on customized domain-specific hardware. Furthermore, we implemented the LUT in software using inline assembly code and filled the task-PE assignments using the profiling information of domain applications on the target hardware.

To support the evaluation of DAS for the EDP objective, the measuring power on the hardware platform at runtime is crucial. The ZCU102 FPGA integrates several onboard current and voltage sensors for the different power rails on the board [44]. These per-rail sensors allow us to accurately measure the power of the processing system (which

contains the CPU cores) and the programmable fabric (which includes the hardware accelerators). Sysfs interfaces in the Linux operating system enable users to read data from these sensors [45]. To this end, we integrated functions that read the sysfs power values into CEDR. CEDR invoked these functions at runtime to accurately measure power consumption and thereby, the EDP.

**Training Setup:** We utilized four real-world applications from the telecommunication and radar domains—WiFi-TX, Temporal Mitigation, Range Detection, and Pulse Doppler—to generate the training data for the DAS preselection classifier. Fifteen different workloads were generated from these four applications by varying the constitution of the number of jobs and their injection intervals to represent a variety of data rates. For example, one workload has 80 Temporal Mitigation and 20 WiFi-TX instances, whereas another one has 100 Range Detection instances. Details of the workload combinations are provided in Appendix C. Each workload was run in streaming mode and repeated for a hundred trials of twelve data points on the FPGA to mitigate runtime variations due to the operating system and memory. Consequently, each data point in Figure 8 represents approximately 150,000 scheduled tasks, with 100 jobs per trial and an average of 15 tasks per job using a specific scheduler. We utilized the same subset of performance counters described in Section 4.2 on the hardware platform to train the DAS preselection classifier model. The model employed a decision tree classifier with a maximum depth of 2. The choice of decision trees as the machine learning technique for the DAS preselection classifier and the tree depth was discussed in Section 4.2. It achieved a classification accuracy of 82.02% when choosing between the slow and fast schedulers at runtime. The accuracy on the hardware platform was lower than observed on the system-level simulator (85.48%) due to runtime variations of the operating system and memory.

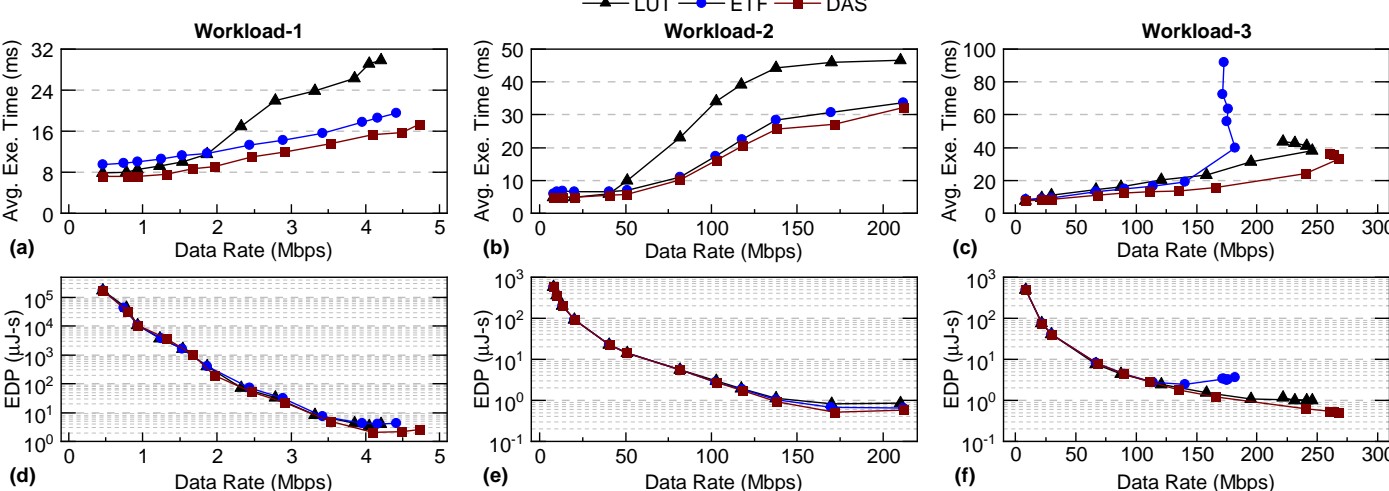

**Figure 8.** A comparison of the average execution time (**a**–**c**) and energy-delay product (EDP) (**d**–**f**) of DAS, LUT, and ETF on a hardware platform for three different workloads.

### 5.2. Performance Results

In this section, we compare the average execution time and EDP of the proposed DAS framework with those of the ETF and LUT schedulers running on the DSSoC configuration implemented on the ZCU102 FPGA. We note that the ETF-ideal scheduler was not implemented on the FPGA since achieving zero overheads on a real system is infeasible. Figure 8 shows a comparison of DAS and the LUT and ETF schedulers for three representative workloads. Figure 8a,d, Figure 8b,e and Figure 8c,f illustrate the average execution time and EDP results for a low data rate (Workload 1), moderate data rate (Workload 2), and high data rate (Workload 3), respectively. At the lower data rate for each workload, the system experienced low congestion levels and exhibited simpler scenarios for task scheduling. Therefore, the DAS classifier predominantly chose the LUT (fast) scheduler, resulting in

similar execution times, as shown in Figure 8a–c. As the data rates increased for Workload 1 and Workload 2, the system experienced more congestion, and hence, the scheduling decisions of the LUT were less optimal. DAS therefore switched between the LUT and ETF, trading off the scheduling overheads of the LUT for the optimality of the ETF. Consequently, DAS achieved an execution time that was notably lower than that of the LUT and ETF schedulers for both Workload 1 and Workload 2. Specifically, DAS achieved execution time improvements of up to 35% and 41% compared to the LUT and ETF, respectively, for Workload 1, and 21% and 13% for Workload 2. The DAS framework also reduces the energy-delay product (EDP) by up to 15% for Workload 1 and up to 21% for Workload 2. As the data rate increased for Workload 3, DAS evaluated and selected the better-performing scheduler between the LUT and ETF at runtime. In this scenario, the DAS framework favored the LUT scheduler since the ETF's overhead resulted in a longer execution time due to the substantial number of ready tasks. Specifically, DAS reduced the execution time by up to 35% and 48% and lowered the EDP by up to 52% compared to the LUT and ETF schedulers for Workload 3. These results show that on a real system, the DAS framework dynamically adapts to the runtime conditions, leading to better performance compared to the schedulers it utilizes.

## 6. Conclusions

Task scheduling is a critical aspect of DSSoC designs, as it targets improvements in performance and energy efficiency without losing the flexibility of general-purpose processors. In this paper, we presented the DAS framework, which combines the advantages of a fast, low-overhead and a sophisticated, high-performance scheduler with a larger overhead for heterogeneous SoCs. The DAS framework achieved a performance time as low as 6 ns and energy overhead as low as 4.2 nJ for a wide range of workload scenarios and 65 ns and 27.2 nJ under heavy workload conditions for applications such as wireless communications and radar systems. We also included the theoretical proof of the DAS framework and its validation using a DSSoC simulator. The experimental results on a hardware platform showed that the DAS framework reduced the average execution time per application instance by up to 48% and the energy-delay product by up to 52%. The experimental results on the DSSoC simulator showed a speedup of up to $1.29\times$ and up to a 45% lower EDP under 40 different workloads. The DAS framework paves the way for DSSoCs to leverage their superior potential to enable the peak performance and energy efficiency of applications within their domain.

**Author Contributions:** Conceptualization, A.A.G., R.M., A.A. and U.Y.O.; Formal analysis, A.A.G., R.M., A.A. and U.Y.O.; Methodology, A.A.G., A.K. and U.Y.O.; Software, A.A.G., S.H. and A.K.; Supervision, U.Y.O.; Validation, A.A.G., S.H. and A.K.; Visualization, A.A.G. and A.K.; Writing—original draft, A.A.G. and S.H.; Writing—review and editing, A.A.G., S.H., A.K., R.M., A.A. and U.Y.O. All authors have read and agreed to the published version of the manuscript.

**Funding:** This material is based on research sponsored by the Air Force Research Laboratory (AFRL) and Defense Advanced Research Projects Agency (DARPA) under agreement number FA8650-18-2-7860. The U.S. Government is authorized to reproduce and distribute reprints for Governmental purposes, notwithstanding any copyright notation thereon. The views and conclusions contained herein are those of the authors and should not be interpreted as necessarily representing the official policies or endorsements, either expressed or implied, of the Air Force Research Laboratory (AFRL) and Defense Advanced Research Projects Agency (DARPA) or the U.S. Government.

**Institutional Review Board Statement:** Not applicable.

**Informed Consent Statement:** Not applicable.

**Data Availability Statement:** Publicly available datasets were analyzed in this study. These datasets can be found at [https://github.com/segemena/DS3, (accessed on 1 September 2023)].

**Conflicts of Interest:** The authors declare no conflict of interest.

## Appendix A. Theoretical Proof of the DAS Framework

This section presents the mathematical background to prove that the DAS framework performs better than both of the underlying schedulers. First, Appendix A.1 derives the conditions required to guarantee that DAS outperforms both the slow and fast schedulers. Then, Appendix A.2 shows that these requirements are satisfied for the trained DAS model using extensive simulations.

*Appendix A.1. Necessary Conditions for the Superiority of DAS*

To understand the behavior of DAS and slow and fast schedulers, we divide a representative data rate–execution time plot into four distinct regions. Each region represents a unique scenario of one scheduler performing better than the others, as shown in Figure A1 and denoted by Equations (A1a), (A1b), (A1c) and (A1d). In Region 1, the system experiences very low congestion levels; hence, both DAS and the fast scheduler perform equally well, whereas the slow scheduler performs poorly. Although the decisions of the slow scheduler are optimal, its benefits are outweighed by the overheads during low system utilization. In Region 2, DAS performs better than both schedulers, but the fast scheduler is still better than the slow scheduler. In Region 3, DAS performs better than both schedulers, but in this case, the slow scheduler is superior to the fast scheduler. In Region 4, DAS performs the same as the slow scheduler, and both perform better than the fast scheduler. We note that none of our extensive simulations experienced Region 4. This region represents very high data rates, where *every* decision made by the slow scheduler differs from the fast scheduler, which is practically infeasible to encounter. So, we exclude this region from the proof to simplify the problem. All these four regions are established on the premise that DAS performs better than or equal to the underlying schedulers.

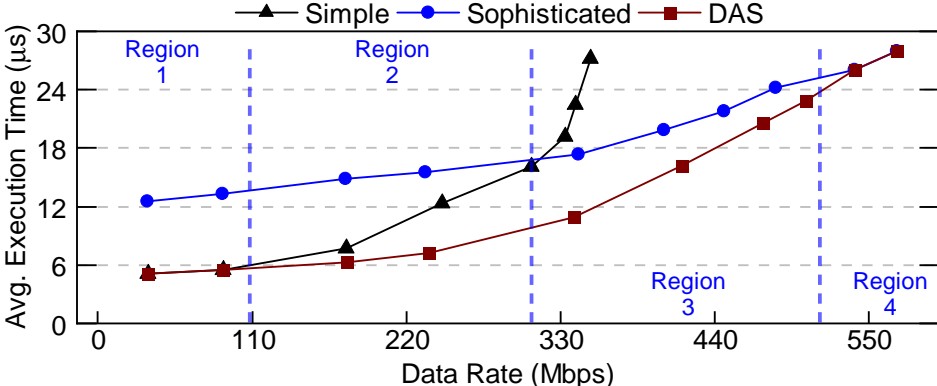

**Figure A1.** Possible regions that cover the comparison between DAS and the fast and slow schedulers.

$$T_{DAS} = T_{FAST} < T_{SLOW} \tag{A1a}$$

$$T_{DAS} = mix(T_{FAST}, T_{SLOW}) < T_{FAST} < T_{SLOW} \tag{A1b}$$

$$T_{DAS} = mix(T_{FAST}, T_{SLOW}) < T_{SLOW} < T_{FAST} \tag{A1c}$$

$$T_{DAS} = T_{SLOW} < T_{FAST} \tag{A1d}$$

We direct our attention solely toward Region 2 due to the following two reasons: (1) Region 1 is a subset of Region 2, and (2) Region 3 is an inverted form of Region 2; hence, we can easily re-derive the criteria by reversing the conditions from Region 2. Furthermore, *it is sufficient to prove that DAS consistently outperforms the fast scheduler, as evidenced by Equation* (A1b). Therefore, we compare the selection of DAS against *the fast scheduler*, while the ideal selection can be *the fast or slow scheduler*.

Table A1 summarizes the notations we use in this section. Specifically, we use $D_{i,ideal}$ to refer to the ideal scheduler selection that will yield optimal performance for task i. On the other hand, $D_{i,DAS}$ represents the scheduler selection made by the DAS preselection classifier. $p_{xy}$ denotes the probability of selecting scheduler X, given that the ideal scheduler

is Y. $\Delta_{i,f}^{xy}$ and $\Delta$ denote the difference in execution time DAS achieves for task i and the entire workload, respectively, with respect to the fast scheduler. We propose the following lemmas to support the fact that DAS is theoretically superior to the underlying schedulers.

**Table A1.** Notations for theoretical proof of DAS scheduler.

| Notation | Definition |
|---|---|
| $D_{i,ideal}$ | Ideal scheduler decision for task i |
| $D_{i,DAS}$ | Decision of DAS scheduler for task i |
| $p_{xy}$ | Selecting scheduler-X when the ideal selection is Y |
| $\Delta_{i,f}^{xy}$ | Execution time difference for task i with respect to the fast scheduler if selecting scheduler X when the ideal selection is Y |
| $\Delta$ | Total execution time difference for all tasks |
| $t_{exe-n}$ | Execution time for the $n$th simulation |

**Lemma A1.** $\Delta_{i,f}^{ff} = \Delta_{i,f}^{fs} = 0$.

**Proof.** In Region 2, we define $\Delta_i$ as the difference in execution time between using DAS and the fast scheduler for Task-i. If DAS always makes the same decisions as the fast scheduler, both schedulers will achieve the same execution time, and hence $\Delta_{i,f}^{ff} = 0$ and $\Delta_{i,f}^{fs} = 0$. $\square$

**Lemma A2.** $\Delta_{i,f}^{ss} < 0$.

**Proof.** Suppose the DAS scheduler selects the slow scheduler when the ideal decision is indeed the slow scheduler. Then, DAS will perform better than the fast scheduler, resulting in a reduction in the execution time, and hence $\Delta_{i,f}^{ss} < 0$. $\square$

**Lemma A3.** $\Delta_{i,f}^{sf} > 0$.

**Proof.** If the DAS scheduler selects the slow scheduler for task i when the ideal decision is the fast scheduler, it will perform poorly because the slow scheduler incurs additional overheads. Consequently, the execution time will be *longer than the fast scheduler*. $\square$

Using the lemmas described above, we can formulate $\Delta_{i,f}$ as follows:

$$\Delta_{i,f} = p_{ff}\Delta_{i,f}^{ff} + p_{fs}\Delta_{i,f}^{fs} + p_{sf}\Delta_{i,f}^{sf} + p_{ss}\Delta_{i,f}^{ss} \tag{A2a}$$

$$\Delta_{i,f} = p_{sf}\Delta_{i,f}^{sf} + p_{ss}\Delta_{i,f}^{ss} \tag{A2b}$$

$$\Delta_{i,f} = p_{sf}\Delta_{i,f}^{L} + p_{ss}\Delta_{i,f}^{G}, \qquad 1 \le i \le N \tag{A2c}$$

where each $p_{xy}\Delta_{i,f}^{xy}$ represents a different combination of DAS selecting scheduler X, given that the ideal scheduler is Y. As $\Delta_{i,f}^{ff}$ and $\Delta_{i,f}^{fs}$ are zero from Lemma A1, we can simplify Equation (A2a) as Equation (A2b). $\Delta_{i,f}^{G}$ and $\Delta_{i,f}^{L}$ denote the gain and loss in the execution time compared to the fast scheduler in Equation (A2c), respectively. Equation (A2) shows the execution time difference for task i. To show the total difference in the execution time, we follow the steps below.

**Definition A1.** *Let $\Delta$ be the total change in the execution time from always choosing the fast scheduler in Region 2.*

$$\Delta = \sum_{i=1}^{N} \Delta_{i,f} = \sum_{i=1}^{N} p_{sf} \Delta_{i,f}^{L} + p_{ss} \Delta_{i,f}^{G} \tag{A3a}$$

$$\Delta = p_{sf} \sum_{i=1}^{N} \Delta_{i,f}^{L} + p_{ss} \sum_{i=1}^{N} \Delta_{i,f}^{G} \tag{A3b}$$

$$\Delta = p_{sf} \Delta^{L} + p_{ss} \Delta^{G} \tag{A3c}$$

The overall change in the execution time, $\Delta$, can be calculated by summing the changes in the execution time for each task, as demonstrated in Equation (A3a). In this equation, the selection probabilities, $p_{sf}$ and $p_{ss}$, are constant values and can be moved outside the summation as common sub-terms. Therefore, we can represent $\Delta$ using Equation (A3b). Then, we define the total gain and loss in the execution time as $\Delta^{G}$ and $\Delta^{L}$ in Equation (A3c).

**Theorem A1.** $\Delta < 0$ *for the DAS scheduler.*

In order to prove that the DAS scheduler is superior to both the fast and slow schedulers, it is necessary to demonstrate that it achieves a lower total execution time. This implies that the overall change in the execution time, denoted by $\Delta$, must be negative.

$$\Delta = p_{sf} \Delta^{L} + p_{ss} \Delta^{G} < 0 \tag{A4a}$$

$$p_{sf} \Delta^{L} < -p_{ss} \Delta^{G} \tag{A4b}$$

$$\text{Since } \Delta^{G} < 0 \text{ and } \Delta^{L} > 0, \quad p_{sf} |\Delta^{L}| < p_{ss} |\Delta^{G}| \tag{A4c}$$

$$\frac{p_{sf}}{p_{ss}} < \frac{|\Delta^{G}|}{|\Delta^{L}|} \tag{A4d}$$

$\Delta^{G}$ is always negative since it denotes the gain in the total execution time. Therefore, we use the absolute value of $\Delta^{G}$ to transform Equation (A4b) into Equation (A4c). In Equation (A4d), we derive the criterion that DAS must achieve to always outperform the underlying schedulers. To ensure superior performance to the fast scheduler in Region 2, the DAS preselection classifier must possess a high value of $p_{ss}$, representing the similarity to ideal decisions, and a low value of $p_{sf}$. Hence, the ratio of $p_{sf}/p_{ss}$ should be low and less than the ratio of $|\Delta^{G}|/|\Delta^{L}|$.

*Appendix A.2. Experimental Validation of the Proof*

We validate the proof by determining the empirical values of the quantities used in Appendix A.1 through a systematic simulation study. This systematic study comprises two steps, which are described next.

Appendix A.2.1. Finding the Empirical Values for $\Delta^{L}$ and $\Delta^{G}$

In order to utilize the equations discussed in Appendix A.1, it is necessary to determine the ideal scheduler decisions and compute the $\Delta^{L}$ and $\Delta^{G}$ values, which are outlined in Algorithm A1. For each ready task $T_i$, the algorithm checks if the decisions of the fast and slow schedulers are identical. If this is the case, the ideal scheduler decision for $T_i$ is the fast scheduler because it incurs a lower overhead for the same decision. If the decisions of the fast and slow schedulers are different, the algorithm employs the decision of the slow scheduler for $T_i$ and the fast scheduler for all other remaining tasks until the end of the simulation, $exe - n$. Following the simulation, the algorithm compares the execution time of the simulation, $t_{exe-n}$, to that of $t_{fast-n}$, which utilizes the decisions of the fast scheduler

for all tasks. If $t_{exe-n}$ is less than $t_{fast-n}$, it implies that the ideal scheduler decision for $T_i$ is the slow scheduler, as it yielded better performance than the fast scheduler. Furthermore, the algorithm adds $t_{exe-n} - t_{fast-n}$ to $\Delta^G$ since it represents the gain in the total execution time. If $t_{exe-n}$ is greater than $t_{fast-n}$, the ideal scheduler decision for $T_i$ is the fast scheduler since utilizing the slow scheduler resulted in a higher execution time. For this scenario, the algorithm adds $t_{exe-n} - t_{fast-n}$ to $\Delta^L$ since it represents the loss in the total execution time.

We used the applications listed in Table 2 for the simulations. The simulations were repeated for ten trials of five data rates to account for runtime variations, scheduling approximately 125,000 tasks each run. After the simulations, we calculated the $|\Delta^G|/|\Delta^L|$ ratio to be 0.793. Therefore, we must demonstrate that a DAS preselection scheduler with a $p_{sf}/p_{ss}$ ratio of less than 0.793 would outperform the fast scheduler.

---

**Algorithm A1** Algorithm to find ideal decisions, $\Delta^L$, and $\Delta^G$ values

---

1: $\mathcal{T}$ = All tasks in the simulation
2: $t_{exe-n}$ = Execution time for the nth simulation
3: $t_{fast-n}$ = Execution time for the nth simulation using the fast scheduler
4: **for** task $T_i \in \mathcal{T}$ **do**
5:     Check $D_{i,fast}$ and $D_{i,slow}$
6:     **if** $D_{i,fast} == D_{i,slow}$ **then**
7:         $D_{i,ideal}$ = Fast
8:     **else**
9:         Use $D_{i,slow}$ for $T_i$, and $D_{i,fast}$ for all following tasks until the end of the simulation $exe - n$
10:         **if** $t_{exe-n} < t_{fast-n}$ **then**
11:             $D_{i,ideal}$ = Slow
12:             $\Delta^G = \Delta^G + t_{exe-n} - t_{fast}$
13:         **else**
14:             $D_{i,ideal}$ = Fast
15:             $\Delta^L = \Delta^L + t_{exe-n} - t_{fast}$
16:         **end if**
17:     **end if**
18: **end for**

---

Appendix A.2.2. Validating the DAS Framework Superiority

We conducted a comprehensive set of simulations to verify the validity of Equation (A4d) for the DAS preselection classifier by varying the $p_{ss}$ and $p_{sf}$ values. We incremented the probability values by 10% in these simulations and performed 121 simulations, with five iterations per $p_{ss}$ and $p_{sf}$ combination. This resulted in approximately 12,500 scheduled tasks per data point. Figure A2 presents the average execution time with an extensive sweep of the parameters $p_{ss}$ and $p_{sf}$. The figure shows an overall trend of an increasing execution time when $p_{sf}$ increases while $p_{ss}$ remains the same. The reason for the increasing execution time is that using a slow scheduler instead of its fast counterpart results in additional overheads. Similarly, when we decrease $p_{ss}$ while keeping $p_{sf}$ the same, we see a trend of an increasing execution time. The plot also includes the results of the slow and fast schedulers as solid lines and the empirically found delta ratio as a dashed vertical line. In the region where the $p_{sf}/p_{ss}$ ratio is less than 0.793, the lines are always in the shaded area. This indicates that whenever the relation in Equation (A4c) holds, the execution time is always better than that achieved with the fast scheduler and, therefore, better than the slow scheduler in Region 2. Thus, for a trained DAS preselection classifier model to guarantee that it outperforms both schedulers, the $p_{sf}/p_{ss}$ ratio must be less than 0.793. We analyzed the performance of the trained DAS preselection classifier model used in Section 4 and included a star-shaped point in Figure A2 to represent it. Our trained DAS preselection classifier model has a $p_{sf}/p_{ss}$ ratio of 0.376. Therefore, our DAS framework is *guaranteed to outperform both schedulers*, as its $p_{sf}/p_{ss}$ ratio is less than the threshold.

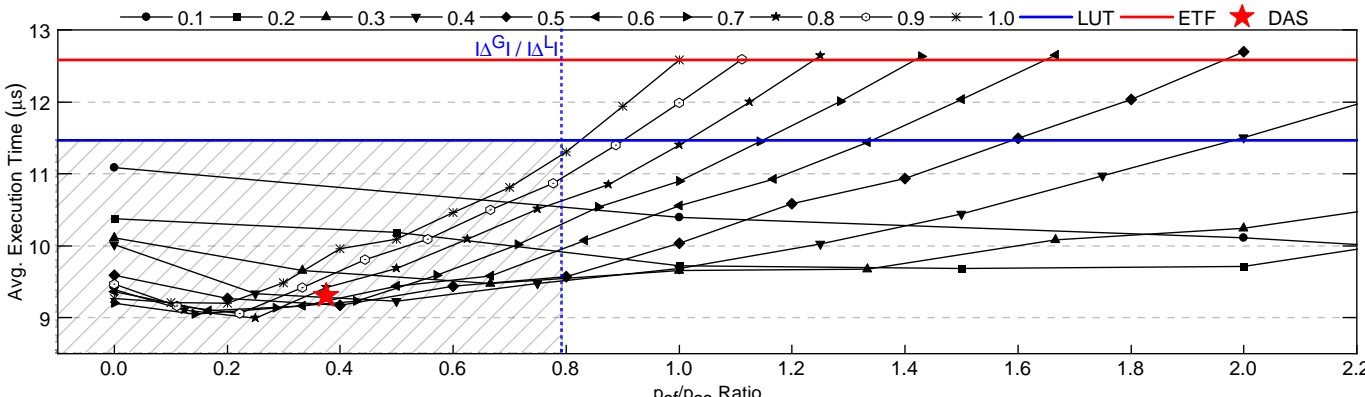

**Figure A2.** Experimental results, showing the average execution times for different $p_{sf}/p_{ss}$ ratios. The results of the fast and slow schedulers are also indicated in the figure by straight lines. The DAS preselection classifier model used in Section 4 is also indicated by a red star. The shaded region is where the expected trained model results should be.

## Appendix B. DSSoC Simulator Workload Mixes

Figure A3 illustrates the workload mixes used in the DSSoC simulator. Each workload is a mix of multiple instances of five applications, consisting of 100 jobs (approximately 140,000 tasks) in total. Each workload is executed at 14 different data rates. The stacked bar for each workload denotes the number of instances of each type of application. For example, Workload 6 (WKL-6) uses 50 instances of WiFi-Transmitter and 50 instances of App-1, and Workload 31 (WKL-31) uses 20 instances of each type. The distribution is selected to represent a variety of low, medium, and high data rates. These workload mixes are used for all the experiments in Section 4.

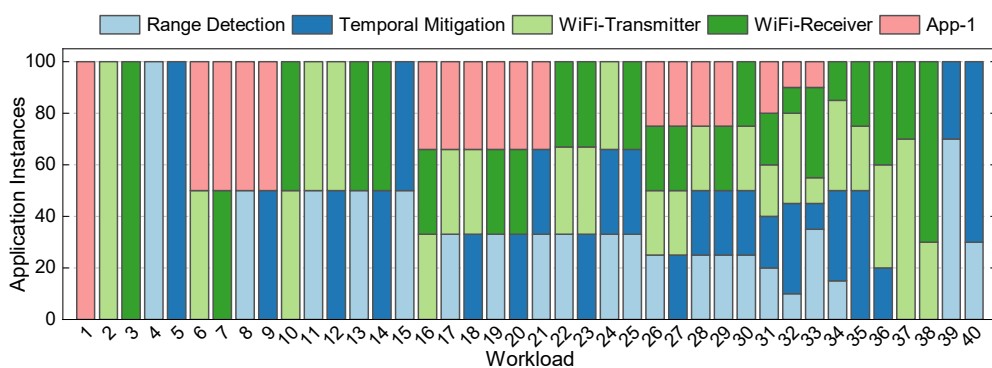

**Figure A3.** The distribution of the number and type of application instances in the 40 workloads used for the evaluation of the DAS framework in the DSSoC simulator.

## Appendix C. Runtime Framework Workload Mixes

Figure A4 presents the workload mixes used in the DSSoC simulator. Each workload consists of 100 jobs, and the stacked bar for each workload denotes the number of instances of each type of application. For example, Workload 7 (WKL-7) uses 80 instances of Temporal Mitigation and 20 instances of WiFi-Transmitter, and Workload 14 (WKL-14) uses 30 instances of Pulse Doppler and 70 instances of WiFi-Transmitter. The distribution is selected so that it represents a variety of low, medium, and high data rates. Also, we try to balance the distribution of applications. For example, the Pulse Doppler application has highly parallel FFT tasks that dominate the system. Therefore, we try to minimize the number of Pulse Doppler instances. Otherwise, the system is overwhelmed by the FFT tasks from a few instances of Pulse Doppler.

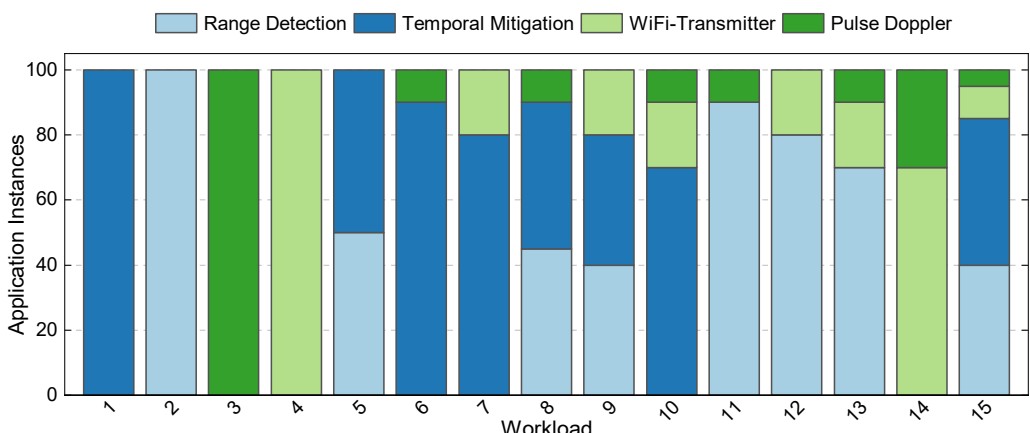

**Figure A4.** The distribution of the number and type of application instances in the 15 workloads used for the evaluation of the DAS framework in the runtime framework.

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
