# Peer review of "Theoretical Validation and Hardware Implementation of Dynamic Adaptive Scheduling for Heterogeneous Systems on Chipâ€"

_jlpea, doi:10.3390/jlpea13040056_

Round 1
Reviewer 1 Report
This is a very interesting and thorough research work targetting task scheduling, a critical area for DSSoC designs.
However, some points should be addressed as well.
The authors refer to the challenge of
"the overheads of switching between policies are not considered as part of the scheduling overhead",
but the performance counters used, or overall flowchart (Fig.2) do not take this into account.
Section 3.3 should elaborate on the method, setup, code and results.
the results appear suddenly: "Experiments show that our fast scheduler takes 236
7.2 cycles (6 ns on Arm Cortex-A53 at 1.2 GHz) on average, and incurs negligible (2.3 nJ) energy 237
overhead."
Is this the result of a simple lookup in a table? this obviously gives superior performance over ETF, but a lot of works do that. On top some are ignored, such as:
i) O. Tomoutzoglou, et al., "Efficient Job Offloading in Heterogeneous Systems Through Hardware-Assisted Packet-Based Dispatching and User-Level Runtime Infrastructure," in IEEE Transactions on Computer-Aided Design of Integrated Circuits and Systems, vol. 39, no. 5, pp. 1017-1030, May 2020, doi: 10.1109/TCAD.2019.2907912.
ii) A. Prodromou, et al., "Agon: A Scalable Competitive Scheduler for Large Heterogeneous Systems, https://arxiv.org/pdf/2109.00665.pdf
(iii) D. Mbakoyiannis, et al., 2018. Energy-Performance Considerations for Data Offloading to FPGA-Based Accelerators Over PCIe. ACM Trans. Archit. Code Optim. 15, 1, Article 14 (March 2018), 24 pages. https://doi.org/10.1145/3180263
The methodology and detailed presentation of theory, calculations and experimental evaluation to result in the performance, power, energy numbers presented are not there, and should be added.
On top, the choice of the classifier should be justified (DT), and also why the classification accuracy of 82% is sufficient.
Last but not least, the article could provide a discussion on how to tackle tasks that are dynamic, and not being able to be profiled offline.
For example, from another perspective, the interactions of accelerated tasks due to sharing of resources (e.g. memory, cache, NoC) can hardly be profiled, as also addressed in (iii) above.
Reviewer 2 Report
The goal of this paper, as exposed by the authors, is to propose a DAS framework that combines the advantages of a fast, low-overhead scheduler and a sophisticated, high-performance scheduler with a larger overhead.
The authors successfully quantify the main contribution in the abstract, introduction and conclusion sections. For each point mentioned in the contribution paragraph, located in chapter 1, authors identify which part that point: [6], [13], [14]. The author should clarify the scientific contributions made in THIS paper.
Section 1 and 2 contains a well-structured introduction and Related Work chapter. The articles in the related works ([15]-[35]) are relatively new and important in the field addressed. A comparison with other projects presented in section 2 is necessary to qualitatively increase the level of the article and the contribution made in the field. The paper present in detail the problems, current limitations and challenges of researchers regarding the task scheduling, DSSoC, and runtime classification.
The results are sufficiently presented, successfully presenting the obtained data from table 4 in relation to Figure 3. Algorithm 1 is not textual presented as much as it should. What datasets or benchmarks were used to test Algorithm 1? Are LUT values obtained after post synthesis or post implementation stages? Is the performance better on the DAS framework with ETFs compared to a HW-RTOS? (like https://www.renesas.com/us/en/software-tool/hw-rtos)
The reference section is good, citing new and relevant articles in the research area.
Round 2
Reviewer 1 Report
All concerns are sufficiently addressed! excellent work.
Reviewer 2 Report
The paper was improved by the revision process.